# Outflows from the youngest stars are mostly molecular

T. P. Ray[1,2 ✉], M. J. McCaughrean[3], A. Caratti o Garatti[4], P. J. Kavanagh[1,5], K. Justtanont[6], E. F. van Dishoeck[7], M. Reitsma[3,7], H. Beuther[8], L. Francis[7], C. Gieser[9], P. Klaassen[10], G. Perotti[8], L. Tychoniec[11], M. van Gelder[7], L. Colina[12], Th. R. Greve[13], M. Güdel[8,14,15], Th. Henning[8], P. O. Lagage[16], G. Östlin[17], B. Vandenbussche[18], C. Waelkens[18] & G. Wright[10]

The formation of stars and planets is accompanied not only by the build-up of matter, namely accretion, but also by its expulsion in the form of highly supersonic jets that can stretch for several parsecs[1,2]. As accretion and jet activity are correlated and because young stars acquire most of their mass rapidly early on, the most powerful jets are associated with the youngest protostars[3]. This period, however, coincides with the time when the protostar and its surroundings are hidden behind many magnitudes of visual extinction. Millimetre interferometers can probe this stage but only for the coolest components[3]. No information is provided on the hottest (greater than 1,000 K) constituents of the jet, that is, the atomic, ionized and high-temperature molecular gases that are thought to make up the jet's backbone. Detecting such a spine relies on observing in the infrared that can penetrate through the shroud of dust. Here we report near-infrared observations of Herbig-Haro 211 from the James Webb Space Telescope, an outflow from an analogue of our Sun when it was, at most, a few times $10^4$ years old. These observations reveal copious emission from hot molecules, explaining the origin of the 'green fuzzies'[4–7] discovered nearly two decades ago by the Spitzer Space Telescope[8]. This outflow is found to be propagating slowly in comparison to its more evolved counterparts and, surprisingly, almost no trace of atomic or ionized emission is seen, suggesting its spine is almost purely molecular.

Herbig-Haro 211 (hereafter HH 211) lies at a distance of $321 \pm 10$ pc (ref. 9) within the Perseus Molecular Cloud. It is one of the most embedded, youngest and nearest protostellar outflows[10,11], making it an ideal target for the high-resolution and penetrating infrared capabilities of the James Webb Space Telescope (JWST). Earlier observations with ground-based telescopes revealed giant redshifted (northwest) and blueshifted (southeast) bow shocks and cavity-like structures in shocked $H_2$ (refs. 10,12) and CO millimetre emission lines[13] respectively, as well as a knotty and wiggling bipolar jet in SiO[14]. At its core is HH 211-mm, a Class 0 protostar that is currently only 0.08 $M_\odot$ in mass[15] but in the process of accreting from a 0.2 $M_\odot$ torus of gas and dust surrounding it[13].

Our Cycle 1 Guaranteed Time Observation programme with the National Aeronautics and Space Administration (NASA)/European Space Agency (ESA)/Canadian Space Agency (CSA) JWST (program identification (PID) 1257, principal investigator (PI) T. P. Ray) consisted of near-infrared camera (NIRCam)[16] imaging of the whole of HH 211,

as well as near-infrared spectrograph (NIRSpec[17]) and mid-infrared instrument (MIRI[18]) integral field spectroscopy of the southeast lobe. Preliminary findings from our NIRCam and NIRSpec observations, which were executed first, are presented here.

A three-colour NIRCam image (Fig. 1) shows a series of bow shocks to the southeast (lower-left) and northwest (upper-right) as well as the narrow bipolar jet that powers them in unprecedented detail, at roughly 5–10 times higher spatial resolution than any previous images. Overall, the emission is red, that is, coming from the F470N filter in the colour composite and hence dominated by the 4.69 µm $H_2$ 0–0 S(9) line. Additional $H_2$ emission is present (for example, the much fainter 1–0 O(5) transition), as well as contributions from other molecules, such as fundamental rotational–vibrational emission from CO (see below and Methods).

The inner jet is seen to wiggle with mirror symmetry on either side of the central protostar. This is in agreement with observations on smaller scales in SiO (ref. 14) and suggests that the protostar may be

[1]Dublin Institute for Advanced Studies, Dublin, Ireland. [2]School of Physics, Trinity College Dublin, Dublin, Ireland. [3]European Space Agency, ESTEC, Noordwijk, the Netherlands. [4]INAF - Osservatorio Astronomico di Capodimonte, Naples, Italy. [5]Department of Experimental Physics, Maynooth University, Maynooth, Ireland. [6]Department of Space, Earth and Environment, Chalmers University of Technology, Onsala Space Observatory, Onsala, Sweden. [7]Leiden Observatory, Leiden University, Leiden, the Netherlands. [8]Max-Planck-Institut für Astronomie (MPIA), Heidelberg, Germany. [9]Max-Planck-Institut für Extraterrestrische Physik, Garching, Germany. [10]UK Astronomy Technology Centre, Royal Observatory Edinburgh, Edinburgh, UK. [11]European Southern Observatory, Garching, Germany. [12]Centro de Astrobiología (CAB, CSIC-INTA), Carretera de Ajalvir, Torrejón de Ardoz, Spain. [13]DTU Space, Technical University of Denmark, Kongens Lyngby, Denmark. [14]Department of Astrophysics, University of Vienna, Vienna, Austria. [15]ETH Zürich, Institute for Particle Physics and Astrophysics, Zurich, Switzerland. [16]Université Paris-Saclay, Université Paris Cité, CEA, CNRS, AIM, Gif-sur-Yvette, France. [17]Department of Astronomy, Stockholm University, AlbaNova University Center, Stockholm, Sweden. [18]Institute of Astronomy, KU Leuven, Leuven, Belgium. ✉e-mail: tr@cp.dias.ie

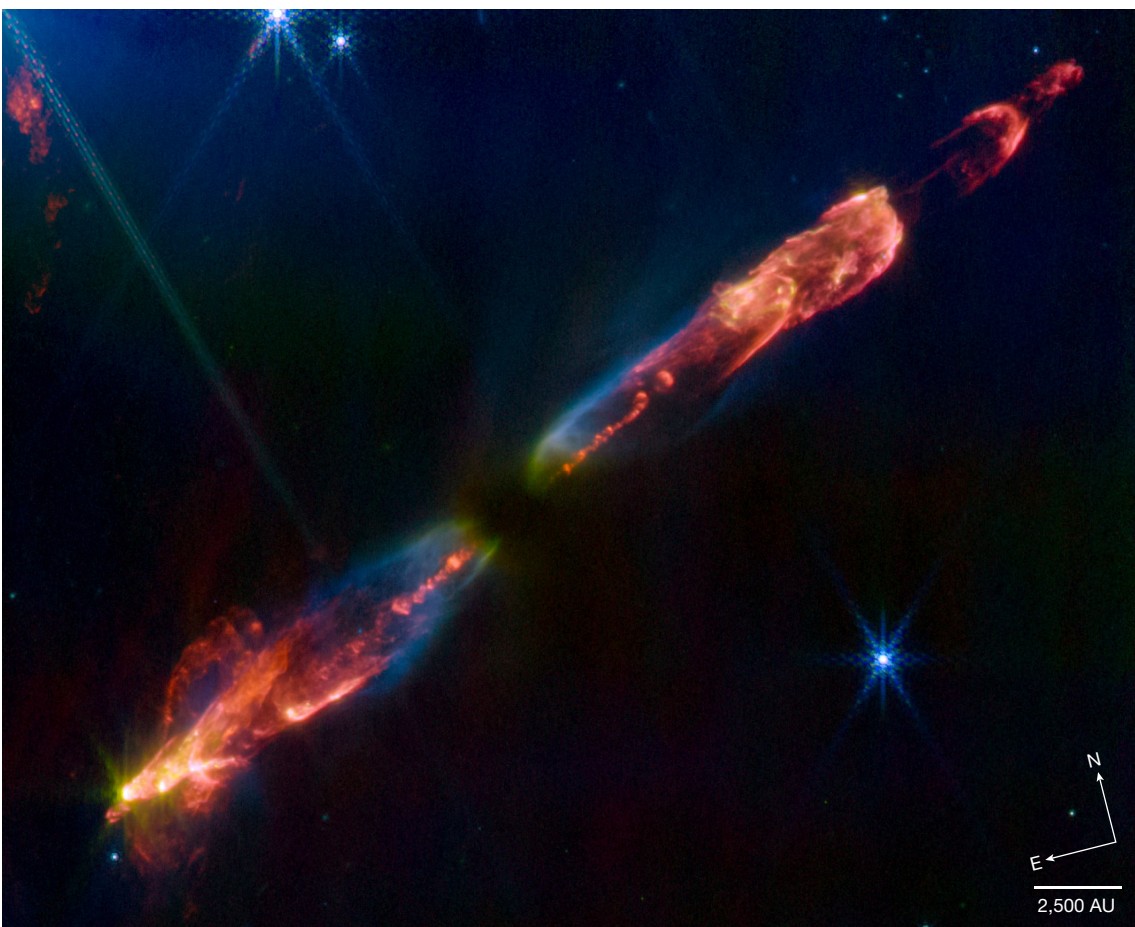

**Fig. 1 | JWST image of HH 211.** A three-colour NIRCam composite of the bipolar HH 211 outflow consisting of images made in the F335M (blue), F460M (green) and F470N (red) filters. The individual images were logarithmically scaled in intensity to compress the dynamic range before being combined. The northwest portion of the outflow is known to be redshifted and the southeast blueshifted. The medium-band F335M filter includes several $H_2$ lines, most notably 3.23 μm $v = 1-0$ O(5), as well as continuum nebulosity. The medium-band F460M filter includes the 4.69 μm 0–0 S(9) line, many lines of CO rotational–vibrational bandhead emission and continuum nebulosity. The narrow-band F470N filter is dominated by the 4.69 μm $H_2$ line. Note the shift to longer wavelengths in scattered light with proximity to the protostar that lies at the centre of the dark lane. The absence of red light towards the protostar can be attributed to the narrowness of the F470N filter. The orientation and projected scale in astronomical units (AU) assuming a distance of 321 pc are shown in the lower right corner.

an unresolved binary[19]. While the protostar itself is still too embedded to be detected at these wavelengths, it is likely illuminating the conical edges of the outflow cavities through reflection nebulosity. This mostly blue emission turns greener closer to the protostar due to extinction.

The rich NIRSpec (1.7–5.1 μm) spectrum of the apex of the southeast bow shock (Fig. 2) shows many lines from the $H_2$ molecule, including pure rotational transitions (labelled in blue), and rotational–vibrational lines from the $v = 1$ (black), $v = 2$ (red) and higher (unlabelled) vibration levels. A striking feature of the spectrum is the many bright CO (1–0) fundamental R- and P-branch transitions between 4.3 and 5.1 μm interspersed with lines from the corresponding isotopologue $^{13}CO$ (Fig. 2, inset). Faint CO lines between 2.29 and 2.4 μm (from CO overtone $v = 2-0$, 3–1 and 4–2) are also detected along with some weak [FeII] emission (not labelled). Notably, virtually no ionized or atomic transitions are seen, suggesting the outflow is almost entirely molecular.

It was noted many years ago that embedded outflows imaged using the Infrared Array Camera (IRAC) on the Spitzer Space Telescope[8] showed significant variations in intensity between the broad IRAC bands. In particular it was found that in three-colour images made using blue for IRAC Band 1 (centred at 3.6 μm), green for IRAC Band 2 (4.5 μm) and red for IRAC Band 4 (8.0 μm), outflows were primarily 'green'. The origin of this 'green fuzzy' emission was thought to be shock-excited emission, either from the $H_2$ 0–0 S(9) line at 4.69 μm (ref. 20) or the CO (1–0) rotational–vibrational lines that stretch from 4.3 to 5.2 μm (refs. 4–6).

An alternative explanation for the excess 'green' emission was a mixture of $H_2$ emission and scattered continuum from the embedded young star[7]. Regardless of its origin, however, the detection of such 'green' emission[21] was considered a highly effective way of tracing outflows from low and even massive embedded young stars[22], not only with Spitzer but also with the subsequent Wide-Field Infrared Survey Explorer (WISE) mission[23].

Our combined analysis of the NIRCam and NIRSpec observations of HH 211 now clearly demonstrates that the significant excess seen in Spitzer IRAC Band 2 (4.5 μm) is largely due to CO fundamental rotational–vibrational emission.

Figure 3 shows a pair of broadly comparable three-colour composite images of HH 211. The rightmost is the Spitzer IRAC Band 1, Band 2 and Band 4 composite, with the jet clearly green. The leftmost shows the result of substituting the JWST F335M image for Spitzer Band 1 as blue, that is, 3.35 μm for 3.6 μm, and JWST F460M for Spitzer Band 2 as green, that is, 4.6 μm for 4.5 μm, with IRAC Band 4 again as the red image. Despite the wider IRAC filter bandwidths and the dramatically

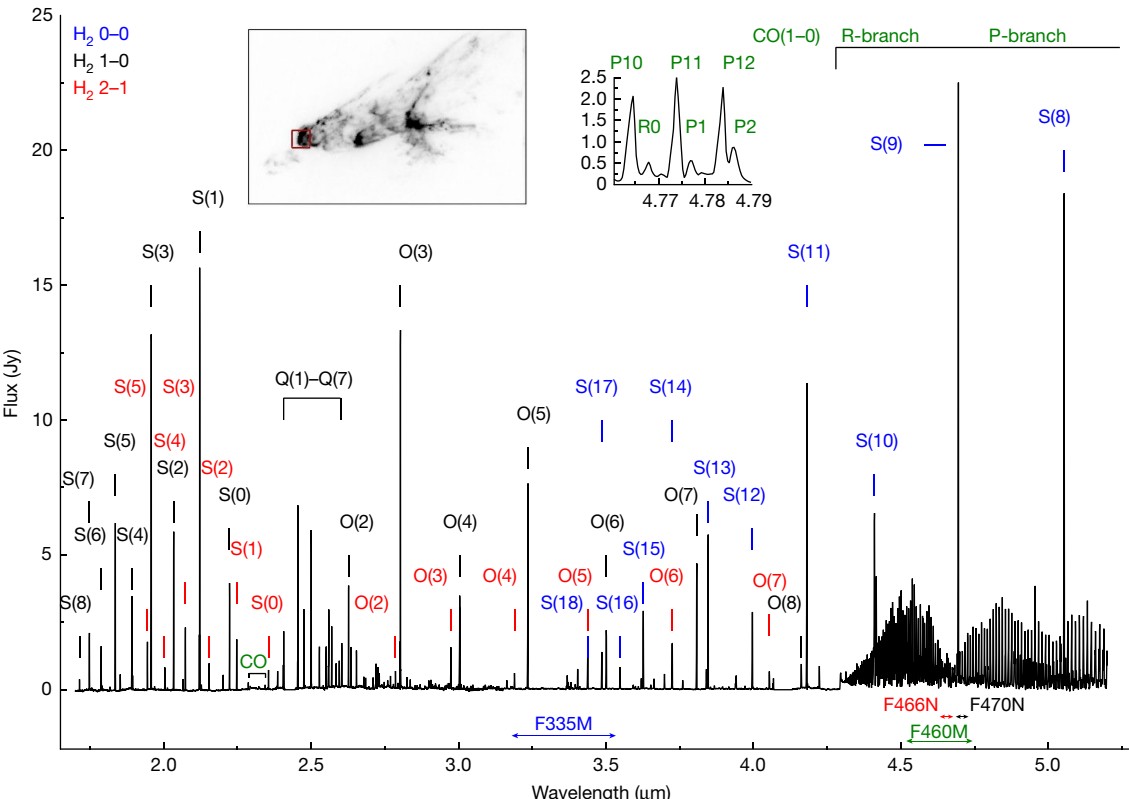

**Fig. 2 | NIRSpec spectrum of the apex of the HH 211 southeast bow shock.** This is an average over the 3″ × 3″ field-of-view of the integral field unit shown in red in the inset image. The most prominent $H_2$ 0–0, 1–0 and 2–1 transitions are shown in blue, black and red, respectively, along with CO transitions in green. Note the presence of the secondary peaks accompanying the $^{12}$CO fundamental series (inset plot) due to the $^{13}$CO isotopologue. The spectral coverage of the NIRCam imaging filters is shown above the wavelength axis for reference.

different spatial resolutions of the telescopes, the resulting colours (derived using the same intensity scale settings) are the same.

To better locate the CO (1–0) emission along the flow and its strength with respect to the brightest $H_2$ line in the F460M filter (the 0–0 S(9) line at 4.69 μm), a 'CO–$H_2$' colour image (Fig. 4 and Methods) shows where CO flux dominates (positive; light yellow-white) and, conversely, where $H_2$ emission prevails (negative; dark orange-red). The CO emission is brightest near the apex of the southeast bow shock, and in a few locations within the northwest bow shocks, whereas the $H_2$ emission is clearly brightest along the jet and within the bulk of the bow shocks. Note that the positive fluxes towards the cavity walls closely

traces F335M (blue) emission seen at the same location in Fig. 1 and is therefore largely scattered light from the protostar rather than pure CO emission. The presence of strong hot CO emission (temperature $T$ greater than 1,000 K) in bow shocks has been predicted for many years[24] and it may even form in outflows from Class 0 sources[25]. Here, however, hot CO is only seen in the bow shocks, not the inner jets, so we may be seeing shocked ambient molecular cloud gas.

The velocities of outflows from young stars are known to vary considerably as a function of the depth of the protostellar gravitational well. In particular, young, highly embedded objects that have only accreted a small fraction of their mass have a shallower gravitational well from

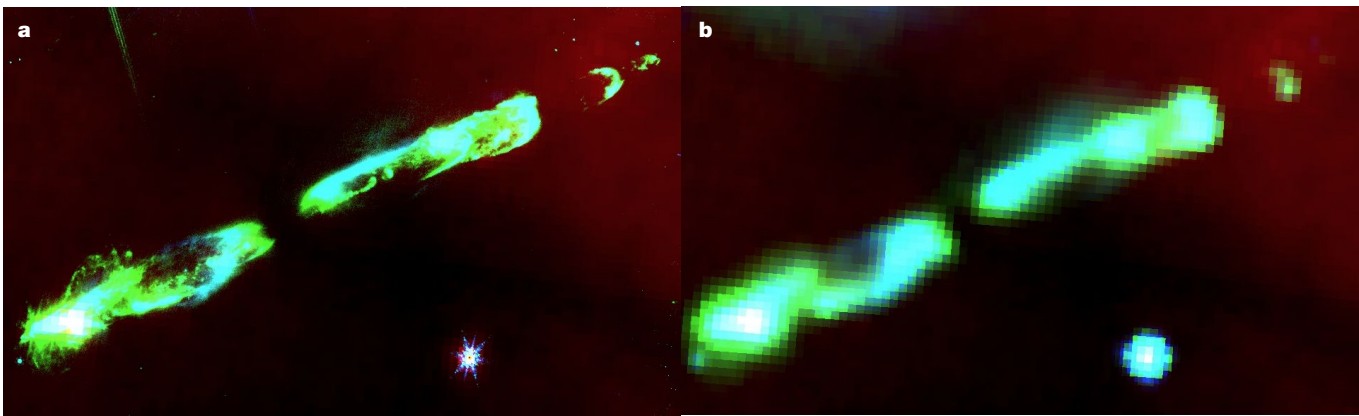

**Fig. 3 | Matching the colours of the 'green fuzzies' in HH 211. a,** Combination of JWST NIRCam F335M (blue), F460M (green) and Spitzer IRAC Band 4 (red) images. **b,** Equivalent combination of Spitzer Band 1, Band 2 and Band 4. Despite the different bandwidths of the filter sets and the different spatial resolutions of the telescopes, the colours match very well.

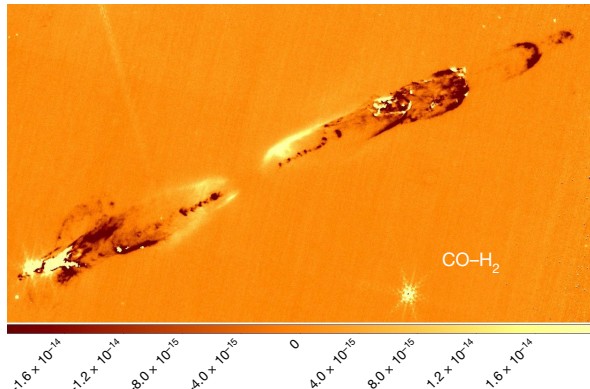

**Fig. 4 | A CO–H₂ image of the HH 211 outflow.** Here increasingly negative fluxes represent increasingly pure H₂ emission and positive fluxes increasingly pure CO emission. For an explanation of how it was produced, see Methods. The colour bar displays line flux densities in erg s⁻¹ cm⁻² arcsec⁻². Note that the positive fluxes along the cavity walls is likely due to scattered continuum emission from the protostar.

which to extract energy and may have much lower outflow velocities than their more evolved counterparts[26].

As the driving source of HH 211 is believed to be extremely young, it may be instructive then to examine the velocities in the outflow. We have done so by comparing our NIRCam F212N image, dominated by the 2.12 μm $v = 1–0$ S(1) line of H₂, with a lower spatial resolution image taken with the European Organisation for Astronomical Research in the Southern Hemisphere (ESO) Very Large Telescope (VLT) Infrared Spectrometer and Array Camera (ISAAC) instrument through a 2.12 μm filter in 2002[12]. Despite the difference in spatial resolution, the baseline of 20 years ensures good proper motion measurements and thus estimates of tangential velocities. We note that the latter are approximately the same as the full three-dimensional velocities because the inclination angle of the outflow is close to the plane of the sky[14].

Figure 5 shows the derived tangential velocity vectors using a cross-correlation technique (Methods) and the previously mentioned assumed distance to HH 211 (ref. 9). The measured velocities are typically around 80–100 km s⁻¹ for the innermost (less than 20″ from the protostar) structures but there is a noticeable slowdown in velocity particularly to the northwest, namely, the redshifted bow shocks. Supplementary Video 1 shows the observed proper motions of HH 211.

Velocities of approximately 100 km s⁻¹ for the inner jet knots are in line with those observed for the even closer-in knots of the SiO flow[14]. As the northwest jet appears to be propagating faster than its associated bow shocks, this suggests the redshifted outflow may be decelerating as it ploughs into its parent cloud. In contrast, the southeast blueshifted jet may be travelling almost ballistically, less impeded by its surroundings. That there is an east-west density asymmetry in the environment is supported by observations (see further discussion of our proper motion results in Methods).

The emission features we see in jets are thought to be due to 'internal working surfaces', that is, shocks where higher-velocity jet material rams into slower material ahead of it. Thus it is the velocity difference between these components, perhaps as little as a few tens of km s⁻¹, that determines the shock velocity[27] rather than any absolute velocity. It follows that the observed jet tangential velocities of around 100 km s⁻¹ may not be the actual shock velocities. Moreover, such high velocity shocks would rapidly dissociate the molecules we are observing[28] and produce an abundance of ionized and atomic species which we do not observe. It is likely therefore that the shocks present in the HH 211 outflow are of low-velocity. An interesting attribute of such shocks is their reduced capacity to destroy dust grains[29]. This ensures

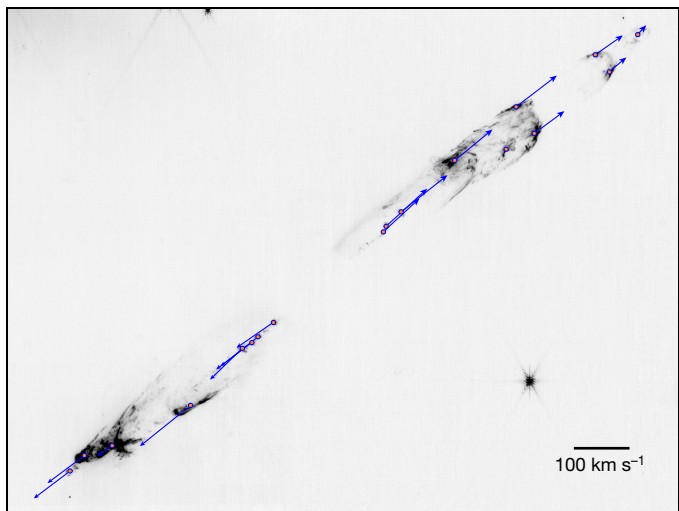

**Fig. 5 | Derived tangential velocities for the 2.12 μm H₂ emission of HH 211.** This uses an ESO VLT image from 2002 in combination with our NIRCam F212N image. Note the systematic decrease in velocity with distance from the protostar for the northwest flow. The plotted points lie at the centre of a region over which the proper motions were determined using a cross-correlation technique (Methods). Errors (1$\sigma$) in the position angles of the proper motion vectors determined from our study are approximately ±6° assuming an average speed of 100 km s⁻¹. For knots moving faster, the position angle error is reduced, and for those moving slower, it is increased. At the centre, but not seen here, is HH 211-mm (right ascension $\alpha$ = 03 h 43 min 56.8054 s; declination $\delta$ = 32°00′ 50.189″, (J2000.0)).

that outflows from very young protostars like HH 211-mm are not just returning gas to the interstellar medium but also dust that has been processed through a protostellar disk. Dust certainly is at least partially destroyed in more evolved outflows[30].

Finally, we note that an important conclusion of this work, largely from our NIRSpec data, is that jets from the earliest protostars are primarily slowly moving molecular beams in contrast to what is found for their more evolved counterparts, such as classical T Tauri stars, where faster-moving (typically greater than 200 km s⁻¹) atomic and ionized species are present in abundance[30].

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

## Methods

### Observations and data reduction

The entire HH 211 field was observed with the JWST NIRCam and a selected region of the HH 211 southeast bow shock (delineated in Fig. 2) with the NIRSpec on 28 August 2022 as part of our HH 211 guaranteed time observation programme (PI: Ray, ID 1257). NIRCam[16] covers the red optical and near-infrared wavelength range 0.6–5.0 μm in two simultaneous channels, namely, the short wavelength (SW) channel (0.6–2.3 μm) and the long wavelength (LW) channel (2.4–5.0 μm). Each consists of two modules (A and B) operating in parallel. The field-of-view of each module is 2.2′ × 2.2′ (4.84 square arcminutes area), and these are separated by a gap of approximately 44″. The total field-of-view for each channel is about 9.7 square arcminutes. In the SW channel, gaps of approximately 5″ separate the four detectors that constitute each module. Both modules in each channel operate with the same set of narrow-, medium- and wide-band filters. The imaging resolution for the SW channel is approximately 31 milliarcsec per pixel, while for the LW channel it is approximately 63 milliarcsec per pixel. Our NIRCam observations of the HH 211 field used the narrow- and medium-band filters listed in Extended Data Table 1. The table includes basic characterization of the filters, the emission lines they cover and associated spatial resolution. In some cases their potential use for continuum subtraction purposes was also considered. All exposures were taken using the BRIGHT1 readout pattern with one integration of eight groups for all short/long filter combinations except F210M/F460M and F162M + F150W2/F335M for which one integration of four groups was used. We took four dithered exposures for each of the long/short filter combinations in the INTRAMODULEX pattern using the STANDARD subpixel dither type.

All NIRCam data was reduced using the JWST pipeline v.1.7.2 (ref. 31), with v.11.16.10 and the version 'jwst_0988.pmap' of the Calibration Reference Data System (CRDS) and CRDS context, respectively. We initially processed all uncalibrated ramp files through Detector1Pipeline along with Image2Pipeline before combining each of the dithers associated with a particular filter into a mosaic using the ImagePipeline.

However, there were two issues with the mosaics produced. As expected our pipeline-reduced images suffered to varying degrees from $1/f$ banding noise[32]. This effect is more evident in narrow-band than medium-band images, as the former have lower background levels. In particular, the NIRCam readout system causes spatial correlations of noise primarily in the fast-read direction, that is, along detector rows, giving rise to the banding. To reduce the $1/f$ noise component we re-ran the Detector1Pipeline with an additional correction based on the ROEBA tool which is an implementation of the row-by-row, amplifier-by-amplifier correction method[32]. However, given the angular scale of HH 211, extreme care was taken to ensure that pixels covered by the extended emission were omitted when determining the median $1/f$ noise used by the correction. This prevented oversubtraction of real extended emission. In some cases, the extended emission covered an entire row or rows of an amplifier so the row-by-row, amplifier-by-amplifier correction was not possible. In these cases we reverted to row-by-row correction only. We reprocessed these new calibrated slope files through Image2Pipeline and Image3Pipeline to reproduce the mosaics, this time with significantly less banding. Additional band denoising was done on the final images using G'MIC-Qt in GIMP before combining the images to make a colour composite.

The second issue encountered was slight subpixel offsets in the relative astrometry between the filter mosaics. To account for these, we determined the sky coordinates for a number of moderately bright stars in the field by fitting their point spread functions with two-dimensional Gaussians. We used their centroids to determine and apply offset corrections for all mosaics relative to the F460M filter.

Note that the images used as blue, green and red in Fig. 1 correspond to the F335M (medium-band filter centred at 3.365 μm and encompassing several H₂ lines, including the 1–0 O(5) transition at 3.323 μm), F460M (medium-band filter centred at 4.624 μm encompassing CO fundamental emission and the H₂ 0–0 S(9) transition) and F470N (narrow-band filter centred on the H₂ 0–0 S(9) transition at 4.695 μm), respectively. It is worth pointing out that only a small portion of the full NIRCam module A and B field is displayed here and the remainder is found to be crisscrossed with several outflows from other embedded protostars in the vicinity including, for example, two almost parallel outflows from IC348 MMS and an additional north-south outflow that appears to be centred on the HH 211 southeast bow shock. The latter may be relevant to proper motion studies described in detail below.

For the NIRSpec observations, the integral field unit (field-of-view 3″ × 3″) was used with the G235H/F170LP (1.66–3.05 μm; $\lambda/\Delta\lambda = 2,700$) and G295H/F290LP (2.87–5.14 μm; $\lambda/\Delta\lambda = 2,700$) grisms to map an approximate 8″ × 11″ region encompassing the southeast bow shock of HH 211 for a total on-source exposure time of 6,336 s (inset in Fig. 2). Two-point nodding was applied as well as a 'leakcal' observations for the micro-shutter assembly leakage correction. Data were reduced and calibrated using the JWST pipeline v.1.8.2 and retrieved from the Mikulski Archive for Space Telescopes. Spectra were extracted and analysed using the Common Astronomy Software Applications[33] and Image Reduction and Analysis Facility[34].

The flux-calibrated spectra presented in Fig. 2 were based on a 1″ × 1″ region centred on the outermost southeast bow shock ($\alpha$ = 3 h 43 min 59. 9 s; $\delta$ = 32°00′ 34.29″, (J2000.0)) marked with a red box in the inset image of Fig. 2.

To produce the CO versus H₂ image shown in Fig. 4 we subtracted the flux-calibrated F470N (H₂) image twice from the corresponding flux-calibrated F460M (CO + H₂) image, thereby ensuring H₂ emission has negative fluxes. We note in passing that the obvious alternative measure of CO emission would be the F466N filter image (Extended Data Table 1), but that encompasses only a few faint CO low-J rotational R-branch lines and thus provides a very limited estimate of the total CO flux. As a check, we also derived the CO/H₂ line flux ratio over the F460M spectral range from our NIRSpec data and compared it with the same ratio at the same location, that is, the bow shock apex, as determined from the NIRCAM imaging. The derived values were 4.33 ± 0.09 and 4.5 ± 0.3 respectively, thus agreeing within the uncertainties.

### Proper motion and tangential velocity measurements

Measuring the proper motion of structures in the HH 211 outflow requires cross-correlating sections of the outflow. This was found to be much more effective than simply tracking the movement of the brightest pixel with time. Moreover, as there is a major difference in the spatial resolution of the ESO VLT and JWST 2.12 μm images, 0.6″ versus 0.08″ respectively (Supplementary Video 1 and Extended Data Fig. 1), applying a cross-correlation technique is superior to pixel tracking if the higher resolution image is convolved to the resolution of the other. The selection of the regions used for cross-correlation is ultimately constrained by the (relatively) poor resolution of the ESO VLT image: tracing the motions of all the small-scale structure seen in the JWST image must await a second epoch of JWST observations. Extended Data Fig. 2 shows a plot of measured tangential velocity versus distance from the source for comparison with Fig. 5. We note here the apparent slowdown in motion to the northwest which may be related to its environment. The presense of an east-west density asymmetry, with denser gas to the west, is supported by mm-band observations. In particular a high-density filament is seen to the west, that is, red-shifted side of HH 211-mm, in H¹³CO⁺ which is also observed in NH₃ (ref. 13). Moreover, recent NH₃ observations show a clear increase in turbulent motion, namely, large line widths, along the direction of the HH 211 northwest jet, suggesting energy may be deposited there from the outflow[35] consisted with the slowdown. One other number that can be derived from the projected distances divided by the tangential velocities, is an dynamical timescale for the system. The latter

is approximately $10^3$ yrs, which is well within the theoretical age of HH 211-mm of a few $10^4$ yrs as expected.

There are two dominant sources of error for the proper motions. The first is the accuracy of registration between the ESO VLT and JWST images, converted from an error in arcseconds to an equivalent speed in km s$^{-1}$. The same value is used for all proper motions. The other error is determined from the cross-correlation technique, by varying the input parameters such as box size and seeing how much the measured motions change. For some knots this is small, as the knots are clean and symmetric, but for others, the error is larger due to confusion with other structures coming in and out of the cross-correlation aperture. The registration error dominates for most but not all knots. Here it is assumed that the errors can be added in quadrature as shown in Extended Data Fig. 2, but note that even in a worst case scenario where the errors would be added linearly, the trends seen are not altered. Finally, there is a systematic error due to the uncertainty in the distance to HH 211 but it is negligible compared to the other errors.

## Data availability

The NIRCam and NIRSpec reduced data are available from the corresponding author upon request. The standard pipeline-reduced data are available publicly one year after the observing date at the MAST portal (https://mast.stsci.edu/portal/Mashup/Clients/Mast/Portal.html).

## Code availability

We reprocessed the raw NIRCam data with the JWST Calibration Pipeline v.1.7.2 (ref. 31), with v.11.16.10 and the version of 'jwst_0988.pmap' of the Calibration Reference Data System (CRDS) and CRDS context, respectively. The pipeline code is publicly available as a GitHub repository at https://github.com/spacetelescope/jwst, with detailed documentation on version installation, running the pipeline and configuring the CRDS version available at https://jwst-pipeline.readthedocs.io/en/latest/. We used the ROEBA tool to reduce 1/$f$ noise in our NIRCam data. The effect and correction are described in ref. 32 with the code and instructions for use publicly available at https://tshirt.readthedocs.io/en/latest/specific_modules/ROEBA.html. Masking of extended emission was performed with a combination of the Astropy[36] io.fits module and the Astropy Regions package[37]. Both are publicly available with detailed instructions on installation and use available at https://www.astropy.org/ and https://astropy-regions.readthedocs.io/en/stable/, respectively. To correct for subpixel offsets between the mosaics in the NIRCam filters we used a combination of the io.fits module for file parsing and editing and the Photutils[38] centroids.centroid_2dg module for centroid measurement. The latter is publicly available with detailed instructions on installation and use available at https://photutils.readthedocs.io/en/stable/index.html. Image visualizations were performed with SAOImage DS9[39], publicly available at https://sites.google.com/cfa.harvard.edu/saoimageds9. Further processing of the images to make the colour composite used GIMP, Adobe LightRoom and Adobe PhotoShop. Processing of the NIRSpec data used CASA and IRAF tools available at https://casa.nrao.edu/ and https://iraf-community.github.io/, respectively.

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

**Acknowledgements** This programme was carried out using JWST Cycle 1 Guaranteed Time Observations combining allocations from the Mid-Infrared Instrument (MIRI) consortium and M.J.M.'s role as a JWST Interdisciplinary Scientist. In particular, the JOYS (JWST Observations of Young Stars) team would like to thank the entire MIRI European and US consortium. Support from STScI is also appreciated. The following National and International Funding Agencies funded and supported MIRI development: National Aeronautics and Space Administration (NASA); European Space Agency (ESA); the Belgian Science Policy Office (BELSPO); Centre Nationale d'Etudes Spatiales (CNES); Danish National Space Centre; Deutsches Zentrum fur Luftund Raumfahrt (DLR); Enterprise Ireland; Ministerio De Economiá y Competividad; Netherlands Research School for Astronomy (NOVA); Netherlands Organisation for Scientific Research (NWO); UK Science and Technology Facilities Council; Swiss Space Office; Swedish National Space Agency; and the UK Space Agency. T.P.R. acknowledges support from European Research Council (ERC) grant no. 743029 EASY. A.C.G. has been supported by PRIN-INAF MAIN-STREAM 2017 and PRIN-INAF 2019 (STRADE). P.J.K. acknowledges financial support from the Science Foundation Ireland/Irish Research Council Pathway programme under grant no. 21/PATH-S/9360. E.F.v.D. and M.v.G. acknowledge support from ERC grant no. 101019751 MOLDISK, grant no. TOP-1614.001.751 from the Dutch Research Council (NWO) and the Danish National Research Foundation through the Center of Excellence 'InterCat' (DNRF150). K.J. acknowledges support from the Swedish National Space Agency (SNSA). T.H. acknowledges support from ERC grant no. 832428 Origins, L.C. acknowledges support from grant no. PIB2021-127718NB-I00, from the Spanish Ministry of Science and Innovation/ State Agency of Research MCIN/AEI/10.13039/501100011033. M.R. perfomed the proper motion analysis as part of a joint Leiden-ESTEC masters project.

**Author contributions** M.J.M. with P.J.K. performed the reduction of the NIRCam and A.C.G. the NIRSpec data, respectively. T.P.R., M.J.M., A.C.G. and P.J.K. led the writing of the manuscript. T.P.R., M.J.M., A.C.G. and K.J. co-led the MIRI Guaranteed Time Observations project on HH 211 as part of the JOYS programme led by E.F.v.D. M.R. carried out the cross-correlation of the JWST and ESO VLT images to derive the tangential velocities. All authors participated in the development and testing of MIRI and its data reduction, in the discussion of the results and/or commented on the manuscript.

**Competing interets** The authors declare no competing interests.

**Additional information**
**Correspondence and requests for materials** should be addressed to T. P. Ray.

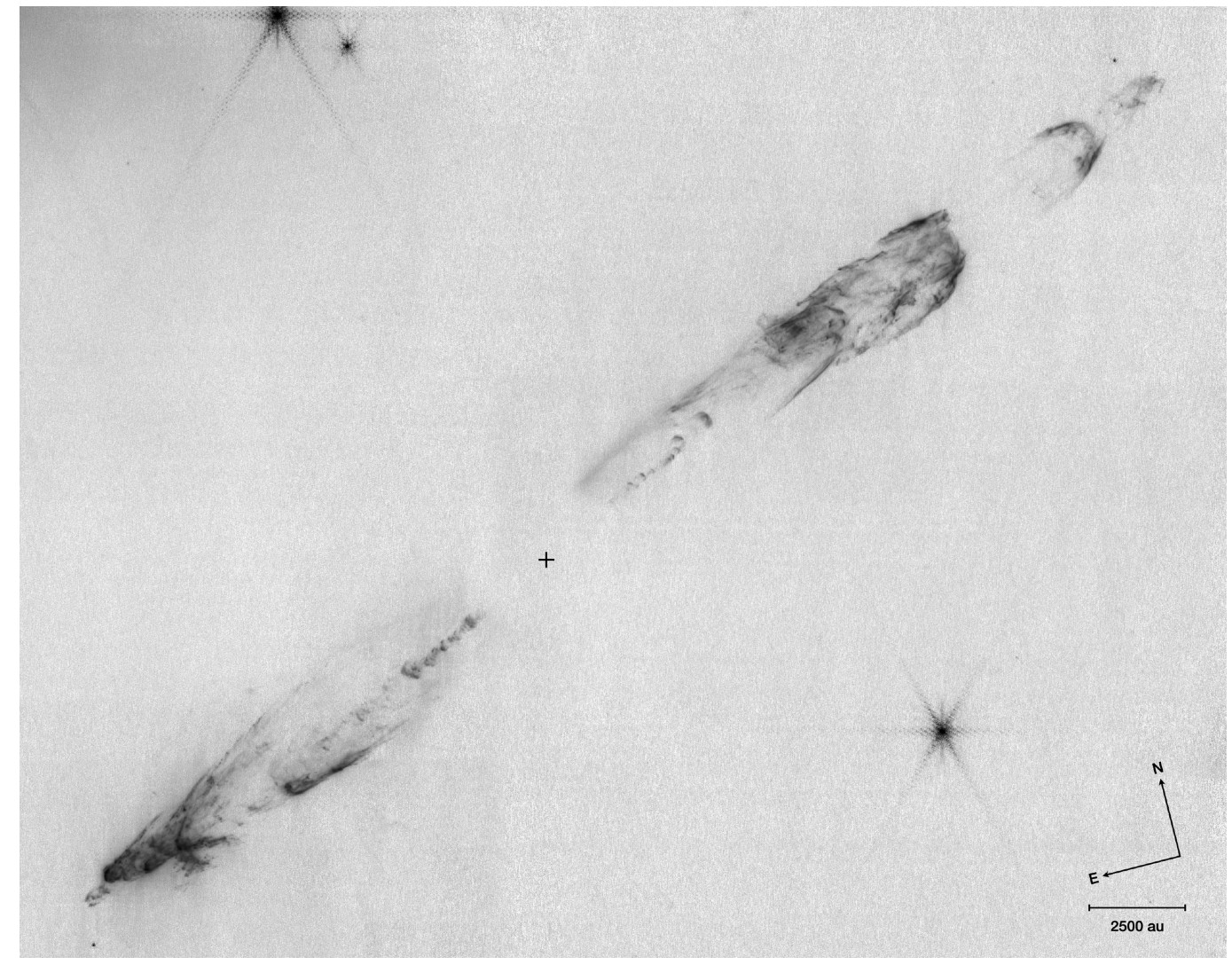

**Extended Data Fig. 1 | JWST NIRCam F212N image of HH 211 with a spatial resolution of 82 milliarcseconds.** The location of HH 211-mm is marked with a cross at the centre of the dark lane. This image was used with the corresponding ESO VLT image from 2002 for proper motion studies.

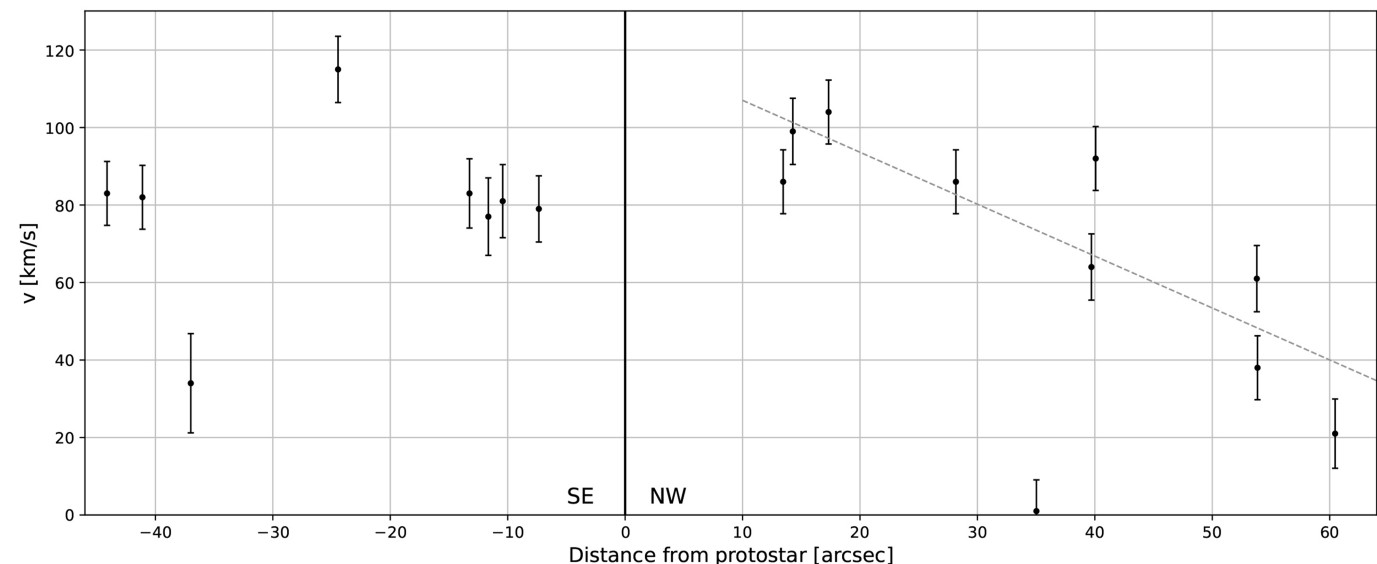

**Extended Data Fig. 2 | A plot of derived tangential outflow velocities versus distance from the protostar.** A clear systematic decrease in velocity with distance is observed in the northwest outflow, whereas the southeast flow has typically higher velocities. Note the dip in tangential velocity at approximately -37″ however, which could be as a result of emission from a crossing north-south outflow seen in the full-scale NIRCam images, that is possibly centred on the HH 211 southeast bow shock. Note error bars are 1$\sigma$.

**Extended Data Table 1 | List of all NIRCAM narrow- and medium-band filters used to image HH 211 and environs**

| Filter | Pivot $\lambda$ ($\mu$m) | Width $\Delta$ ($\mu$m) | Resolution (mas) | Spectral Line(s) | Exposure [a] (s) |
|--------|------------|------------|------------|------------------|--------------|
| F162M | 1.626 | 0.168 | 62.9 | Continuum[b] | 301 |
| F164N | 1.644 | 0.020 | 63.6 | [FeII] | 2×644 |
| F210M | 2.093 | 0.205 | 81.0 | Continuum[b] | 301 |
| F212N | 2.120 | 0.027 | 82.1 | $H_2$ 1–0 S(1) | 644 |
| F323N | 3.237 | 0.038 | 125.3 | $H_2$ 1–0 O(5) | 644 |
| F335M | 3.365 | 0.347 | 130.3 | $H_2$ 1–0 O(5)[c], PAH, $CH_4$ | 301 |
| F460M | 4.624 | 0.228 | 179.0 | $CO$[d], $H_2$ 0–0 S(9) | 301 |
| F466N | 4.654 | 0.054 | 180.2 | $CO$[d] | 644 |
| F470N | 4.707 | 0.051 | 182.2 | $H_2$ 0–0 S(9) | 644 |

a. Here all NIRCAM exposures are listed for reference although only a limited set have been reported on in this paper. Note the total on-source exposure time was 301 sec and 644 sec for the medium- and narrow-band filters, respectively, except for our [FeII] $\lambda1.64\,\mu$m observations which consisted of two co-added 644 sec exposures. b. For modeling the underlying continuum in the [FeII] $\lambda1.64\,\mu$m and $H_2$ 1–0 S(1). c. This hydrogen line is contained within this medium-band filter as well as the F323N filter. d. Fundamental vibration-rotation band of CO.