## [Peer Review File · Nature]

Manuscript Title: The Slow Moving Drama of Stellar Birth as Seen by JWST

Reviewer Comments & Author Rebuttals

Reviewer Reports on the Initial Version:

Referees' comments:

Referee #1 (Remarks to the Author):

Overall: This paper describes early JWST/NIRCam+NIRSpec studies of a famous and spectacular Herbig-Haro object, HH 211. This object has a long history of analysis by major missions, but JWST breaks new ground with resolution and sensitivity in the near infrared, and this paper showcases this with both magnificent spectra and imaging. It is important to publish this result in a timely fashion, as published results in this field are still sparse compared with other key areas in JWST literature. However, this paper showcases a particularly detailed look at the morphology of a Herbig-Haro object, revealing brand new resolution data and some very interesting early results.

Beyond the overall data presentation, the key results include the breaking of the H2 S(9) vs. CO emission line forest degeneracy in the long-debated "Green Fuzzy" IRAC band 2 (4.5 microns) result from the Spitzer era. They do this by cleverly subtracting a narrow band from a medium band observation to generate an index for dominant CO vs. H2. In doing so, they discover spatial stratification between the two species, revealing some interesting physics.

A second key result is a very precise proper motion measurement of the velocities in the flow, complicated by some possible crosswinds on one of the extensions. This data will narrow model constraints on this particular flow, and promises the potential for more accurate modeling in the future.

I have some concerns about the technical language level of the manuscript in the early paragraphs, which seemed almost too "public" level - slightly obscuring some of the scientific points. My comments are relatively minor, revolving around details of the authors' interpretation of the CO emission models, the shock models, and whether the authors have cross-compared their NIRSpec and NIRCam data where overlapping.

Main -

Paragraph 3: What is the name of the protostar? Some basic information on the driving source might be relevant. There are other notable HH objects where the driving source is prominently visible in the flow.

Paragraph 3: Why was the outflow "best seen" in SiO J=8-7? What does "best" mean in this context? Is it because of spatial resolution, brightness in this line, or the instrumental choice?

Figure 1: Suggest (but not require) including the colors in a legend within the figure in a corner. (There are already multiple annotations, so this addition would not affect the figure look.)

Page 3, paragraph 2: "We just note here that the NIRSpec CO spectrum towards the south-eastern bow-shock is consistent with the simulated CO spectrum in Local Thermodynamic Equilibrium (LTE) with $T \sim 2500$ K and $n \sim 10^6$ cm⁻³ as predicted by Tappe and collaborators¹⁷ (see their Fig. 6). While it is certain that the temperature of the CO is $\geq 1,000$ K, we note that if the CO optical depth is high, the gas temperature may be lower than the above LTE value¹⁸."

The text implies that the CO spectrum is consistent with a 2500 K / LTE model, and then immediately challenges this - the CO temperature could be a quite a bit lower and LTE is not necessarily applicable. I am unsure whether the authors have evidence that their data deviates in a significant way from the model (which is a tiny piece of the referenced figure in Tappe+17). I would like the text to clarify whether the authors see a deviation, or whether the uncertainties merely allow a deviation.

Page 6, paragraph 1: "To evaluate the location of the CO (1–0) emission along the flow and its strength with respect to the H2 0-0 S(9) line, an "CO – H2" color image (Figure 4) was created in which areas where CO emission dominates is represented by positive (dark orange) fluxes and those where H2 emission prevails as negative (light yellow) fluxes. This was achieved by subtracting twice the flux calibrated F470N (H2) image from the corresponding flux calibrated F460M (CO + H2) image, thereby ensuring H2 emission has negative fluxes."

The authors also show a NIRSpect IFU spectrum of one of the knots in Figure 2. Does the spatial distribution of the H2 and CO in that small NIRSpect aperture match the full-size "CO-H2" color image in the overlap region? Although the IFU covers only a small region, it seems like it could be verified/registered against the image by, for example, checking whether the border lands in the same pixels. I would like to know if the authors have done this comparison.

Page 8, paragraph 1: " In contrast the south-east blue-shifted jet may be travelling almost ballistically, virtually unimpeded by a less dense surrounding medium."

I'm a bit skeptical that the medium could be so empty with all these young stars around. The authors could refer to something like an extinction map, reddening colors, or other evidence supporting this idea that this region is evacuated, to make their case stronger.

Page 8, paragraph 2: "The observed kinematic velocities of $\approx 100 \text{ km s}^{-1}$ are not the actual shock velocities since such strong shocks would rapidly dissociate the molecules we are observing³³. Instead the usual explanation for the structures we observe is that they are due to higher velocity jet material ramming into slower material ahead of it and thus it is the velocity difference that determines the actual shock velocity³¹."

This argument was difficult to follow, despite several attempts in the text to restate it. If I understand correctly, the authors are arguing that shock arises from the interaction of a very fast moving (call it v_1) jet (measured in proper motion) catching up to a somewhat slower moving (call it v_2) material (previous epoch jet?) but *still moving in the same direction*. They would then argue that the measured shock speed approximately difference between them ($v_1 - v_2$), which is considerably less than v_1 . This slower shock speed is indicated by the detected emission tracers from a relatively slow shock. The faster shock, presumably, might track with the [Fe II] emission in F164N. Regardless, I encourage the authors to clarify this distinction.

Page 8, paragraph 3: "Finally we note that an important conclusion of this work is that the so-called "prime mover", i.e., the underlying jet driving outflows from the earliest protostars, may be largely relatively slowly moving molecular beams in contrast to what is found for their more evolved counterparts such as classical T Tauri stars where faster moving atomic and ionised species dominate in their jets."

Please provide a typical velocity for classical T Tauri star jets, and a supporting citation.

Referee #2 (Remarks to the Author):

Title: The Slow Moving Drama of Stellar Birth as Seen by JWST

Authors: Ray et al ..

This paper presented the new JWST observations of the HH 211 outflow at an angular resolution of $\sim 0.07''$, showing the details of how the HH 211 jet interacts with its surroundings, driving material away. The authors found that ^{12}CO and ^{13}CO can explain the origin of some "Green Fuzzies" detected before. Comparing with the ground-based data taken about 20 yrs ago, the authors found that the outflow is propagating slowly through its surroundings at about 80-100 km/s. Since no trace of atomic or ionized emission is detected in the outflow, the authors suggest that the jet beam coming from the disk should be almost molecular.

Although the JWST images showed the outflow structure in more detail than before, similar scientific results have been reported before. For example, previous observations already showed roughly the same outflow structure in H_2 (Refs 6 and 9) and CO (Ref 10), and the jet beam in H_2 (Ref 6 and 9), CO (Ref 10), and SiO (Refs 9 and 10). Furthermore, the jet beam in those previous CO and SiO observations extends much closer to within 100 au of the protostar, already suggesting more directly that the jet beam or the so-called "prime mover" should be mainly molecular. In addition, previous proper motion measurement also already showed that the jet beam in this source is propagating at ~ 100 km/s (Ref 11), with a slow shock velocities of 20-30 km/s in the jet beam (Ref 11). As a result, I am afraid that the paper here does not have enough significantly new results to be published in Nature.

BTW, in Figure 6, the authors found the tangential velocity of the NW tip to be ~ 20 km/s, only a quarter of that of the SW tip, which is ~ 80 km/s. Since the NW lobe is longer than the SE lobe, and the shape of NW tip is similar to that of the SW tip, I think the velocity of the NW tip should not be that much slower than that of the SW tip. Also, a shock velocity of 20 km/s may not be high enough to produce the observed H_2 emission. So I did a quick measurement by comparing Figure 1 here to Figure 1 in Ref 9, and obtained a tangential velocity of about 45-60 km/s for the NW tip. Could the authors please confirm their measurement?

In Figure 1, the North direction in the upper-left corner is incorrect. The scale length of 1000 au in the lower-right corner should be shorter. It is not clear how the authors pinpoint the location of the central protostar and determine its coordinates. It would be useful to mention the coordinates of the protostar to be clear for the readers.

BTW, I could not find any information about the angular resolution for each filter. May I suggest the authors to list the angular resolution for each filter, e.g., in Table 1?

Author Rebuttals to Initial Comments:

Referees' comments:

Referee #1 (Remarks to the Author):

Overall: This paper describes early JWST/NIRCam+NIRSpec studies of a famous and spectacular Herbig-Haro object, HH 211. This object has a long history of analysis by major missions, but JWST breaks new ground with resolution and sensitivity in the near infrared, and this paper showcases this with both magnificent spectra and imaging. It is important to publish this result in a timely fashion, as published results in this field are still sparse compared with other key areas in JWST literature. However, this paper showcases a particularly detailed look at the morphology of a Herbig-Haro object, revealing brand new resolution data and some very interesting early results.

Beyond the overall data presentation, the key results include the breaking of the H₂ S(9) vs. CO emission line forest degeneracy in the long-debated "Green Fuzzy" IRAC band 2 (4.5 microns) result from the Spitzer era. They do this by cleverly subtracting a narrow band from a medium band observation to generate an index for dominant CO vs. H₂. In doing so, they discover spatial stratification between the two species, revealing some interesting physics.

A second key result is a very precise proper motion measurement of the velocities in the flow, complicated by some possible crosswinds on one of the extensions. This data will narrow model constraints on this particular flow, and promises the potential for more accurate modelling in the future.

I have some concerns about the technical language level of the manuscript in the early paragraphs, which seemed almost too "public" level - slightly obscuring some of the scientific points. My comments are relatively minor, revolving around details of the authors' interpretation of the CO emission models, the shock models, and whether the authors have cross-compared their NIRSpec and NIRCam data where overlapping.

We first of all wish to thank Referee #1 for their very helpful review. We have tightened up considerably on the language used in the first paragraphs as restructuring was necessary to shorten the main text. The language is now certainly more scientific and the content more to the point.

Main -

Paragraph 3: What is the name of the protostar? Some basic information on the driving source might be relevant. There are other notable HH objects where the driving source is prominently visible in the flow.

The name of the protostar, HH212-mm, with some basic information about it, e.g. estimated mass, location, class, etc., is now in the text. Its position is also contained in the caption of

Figure 5 and is marked in Extended Data Figure 1. Note that it can only be observed at long infrared and millimeter wavelengths.

Paragraph 3: Why was the outflow "best seen" in SiO J=8-7? What does "best" mean in this context? Is it because of spatial resolution, brightness in this line, or the instrumental choice?

Actually for all of the above reasons. Until JWST, the best spatial resolution, when investigating embedded protostars, was afforded by ALMA. The SiO J=8-7 line is very bright and observing with a mm interferometer such as ALMA allows one to peer closer to the young star because of the very high optical extinction. To avoid, however, having to explain all of this without expanding the text even further, we decided to just drop the word "best".

Figure 1: Suggest (but not require) including the colors in a legend within the figure in a corner. (There are already multiple annotations, so this addition would not affect the figure look.)

An interesting suggestion but we decided for aesthetic reasons to leave the figure as is because we did not want to add additional annotations.

Page 3, paragraph 2: "We just note here that the NIRSpect CO spectrum towards the southeastern bow-shock is consistent with the simulated CO spectrum in Local Thermodynamic Equilibrium (LTE) with $T \sim 2500$ K and $n \sim 10^6$ cm⁻³ as predicted by Tappe and collaborators¹⁷ (see their Fig. 6). While it is certain that the temperature of the CO is $\geq 1,000$ K, we note that if the CO optical depth is high, the gas temperature may be lower than the above LTE value¹⁸."

The text implies that the CO spectrum is consistent with a 2500 K/LTE model, and then immediately challenges this - the CO temperature could be a quite a bit lower and LTE is not necessarily applicable. I am unsure whether the authors have evidence that their data deviates in a significant way from the model (which is a tiny piece of the referenced figure in Tappe+17). I would like the text to clarify whether the authors see a deviation, or whether the uncertainties merely allow a deviation.

That is a very good point and, as we intend to do more sophisticated modelling of this outflow in the future with Benoit Tabone, with the addition of new Mid-Infrared Instrument (MIRI) data, we have decided to drop these sentences.

Page 6, paragraph 1: "To evaluate the location of the CO (1-0) emission along the flow and its strength with respect to the H₂ 0-0 S(9) line, an "CO - H₂" color image (Figure 4) was created in which areas where CO emission dominates is represented by positive (dark orange) fluxes and those where H₂ emission prevails as negative (light yellow) fluxes. This was achieved by subtracting twice the flux calibrated F470N (H₂) image from the corresponding flux calibrated F460M (CO + H₂) image, thereby ensuring H₂ emission has negative fluxes."

The authors also show a NIRSpect IFU spectrum of one of the knots in Figure 2. Does the spatial distribution of the H₂ and CO in that small NIRSpect aperture match the full-size "CO-H₂" color

image in the overlap region? Although the IFU covers only a small region, it seems like it could be verified/registered against the image by, for example, checking whether the border lands in the same pixels. I would like to know if the authors have done this comparison.

Yes we had verified this. Moreover, in the Methods section we have now added the derived CO/H₂ flux ratio for the same area over the appropriate spectral range using both NIRCam and NIRSpc. Both independently derived ratios agree within the errors.

Page 8, paragraph 1: " In contrast the southeast blue-shifted jet may be travelling almost ballistically, virtually unimpeded by a less dense surrounding medium."

I'm a bit skeptical that the medium could be so empty with all these young stars around. The authors could refer to something like an extinction map, reddening colors, or other evidence supporting this idea that this region is evacuated, to make their case stronger.

This is a good point as there is evidence. We now refer to H¹³CO⁺ and NH₃ observations that show there is an east-west density asymmetry in the immediate environment of HH211-mm. In particular it is clear that the western side contains gas of higher densities and, in addition, there is newer evidence (referenced) that the NW outflow may be driving turbulence in this denser gas, i.e. this outflow is depositing energy. Again this is consistent with the slowing down.

Page 8, paragraph 2: "The observed kinematic velocities of $\approx 100 \text{ km s}^{-1}$ are not the actual shock velocities since such strong shocks would rapidly dissociate the molecules we are observing³³. Instead the usual explanation for the structures we observe is that they are due to higher velocity jet material ramming into slower material ahead of it and thus it is the velocity difference that determines the actual shock velocity³¹."

This argument was difficult to follow, despite several attempts in the text to restate it. If I understand correctly, the authors are arguing that shock arises from the interaction of a very fast moving (call it v1) jet (measured in proper motion) catching up to a somewhat slower moving (call it v2) material (previous epoch jet?) but *still moving in the same direction*. They would then argue that the measured shock speed approximately difference between them (v1 - v2), which is considerably less than v1. This slower shock speed is indicated by the detected emission tracers from a relatively slow shock. The faster shock, presumably, might track with the [Fe II] emission in F164N. Regardless, I encourage the authors to clarify this distinction.

The relevant lines have now be rewritten and hopefully the argument is clearer. The referee is correct in their interpretation of what was meant, so although clumsily written the point was communicated!

Page 8, paragraph 3: "Finally we note that an important conclusion of this work is that the so-called "prime mover", i.e., the underlying jet driving outflows from the earliest protostars, may be largely relatively slowly moving molecular beams in contrast to what is found for their more evolved counterparts such as classical T Tauri stars where faster moving atomic and ionised

species dominate in their jets."

Please provide a typical velocity for classical T Tauri star jets, and a supporting citation.

This has been done.

Referee #2 (Remarks to the Author):

Title: The Slow Moving Drama of Stellar Birth as Seen by JWST

Authors: Ray et al ..

This paper presented the new JWST observations of the HH 211 outflow at an angular resolution of $\sim 0.07''$, showing the details of how the HH 211 jet interacts with its surroundings, driving material away. The authors found that ^{12}CO and ^{13}CO can explain the origin of some "Green Fuzzies" detected before. Comparing with the ground-based data taken about 20 yrs ago, the authors found that the outflow is propagating slowly through its surroundings at about 80-100 km/s. Since no trace of atomic or ionized emission is detected in the outflow, the authors suggest that the jet beam coming from the disk should be almost molecular.

Although the JWST images showed the outflow structure in more detail than before, similar scientific results have been reported before. For example, previous observations already showed roughly the same outflow structure in H_2 (Refs 6 and 9) and CO (Ref 10), and the jet beam in H_2 (Ref 6 and 9), CO (Ref 10), and SiO (Refs 9 and 10). Furthermore, the jet beam in those previous CO and SiO observations extends much closer to within 100 au of the protostar, already suggesting more directly that the jet beam or the so-called "prime mover" should be mainly molecular. In addition, previous proper motion measurement also already showed that the jet beam in this source is propagating at ~ 100 km/s (Ref 11), with a slow shock velocities of 20-30 km/s in the jet beam (Ref 11). As a result, I am afraid that the paper here does not have enough significantly new results to be published in Nature.

First of all we wish to thank Referee # 2 for their very useful comments, which has led us to make more explicit what our new findings are. Obviously the NIRCам composite image of HH 211 shows this outflow in unprecedented detail as acknowledged by Referee # 2. One has only to compare it with the corresponding VLT image (referenced in our paper and shown in the proper motion video in SI) to see the spectacular improvement in resolution and sensitivity afforded by JWST. Equally important, however, is our NIRSspec data. This has approximately 10^3 times better sensitivity than what can be obtained from the ground or even what has been achieved from space to date, albeit at somewhat longer wavelengths, using Spitzer. NIRSspec shows almost a total lack of ionised/atomic species in the band up to 5 micron where we expect to see such lines. Referee # 2 rightly points out that mm interferometers like ALMA show the molecular component of the HH 211 jet, however *that does not prove* the core of the jet is molecular since such interferometers *are only sensitive to molecules*. Instead one has to search for the atomic/ionized

species in the IR band because (a) there are suitable transitions in the band and (b) extinction in the IR is lower than in the optical and we are hampered by extinction for the most embedded protostars.

As regards our proper motion results, yes it was known, and we reference this, that the inner SiO jet knots move at approximately 100 km/s. We show however that the outer knots are moving at a very similar velocity along with the SW bow shocks. This was not known before. Moreover, we also detect a slowing down in the NW bow shock: again a new discovery. Finally we were not claiming to have discovered slow shocks in the HH 211 outflow. All outflows, even the typically faster ones from classical T Tauri stars, or Herbig Ae/Be stars, have some element of slow shocks: even a single bow shock inherently contains a wide range of shock velocities. Instead we were making the point that as so little ionized/atomic species is observed the shocks present must on average be slow (a few tens of km/s at most) and that this in turn suggests dust must be returned to the parent cloud via the outflow without being destroyed. This is not the case with more evolved outflows (we reference this fact). All of these points are now made more clearer in the text.

BTW, in Figure 6, the authors found the tangential velocity of the NW tip to be ~ 20 km/s, only a quarter of that of the SW tip, which is ~ 80 km/s. Since the NW lobe is longer than the SE lobe, and the shape of NW tip is similar to that of the SW tip, I think the velocity of the NW tip should not be that much slower than that of the SW tip. Also, a shock velocity of 20 km/s may not be high enough to produce the observed H₂ emission. So I did a quick measurement by comparing Figure 1 here to Figure 1 in Ref 9, and obtained a tangential velocity of about 45-60 km/s for the NW tip. Could the authors please confirm their measurement?

In light of these remarks, we have also made a “sanity” check of our convolution/cross-correlation results by simply magnifying and rotating the VLT and JWST images until they aligned well (there are enough stars to do that). By comparing the original VLT image to the F212N JWST image and NIRCcam composite in Figure 1 and averaging, we get approximately 20 ± 6 km/s for the velocity of the NW tip. This is not very different to the 21 ± 4 (stochastic) ± 8 (systematic) km/s that we derived using the convolution/cross-correlation technique. In any event it is definitely not 45-60 km/s. In addition, it is worth pointing out that just because a bow shock moves at say 20 km/s in the observer’s frame (i.e. ours), it does not follow that the corresponding shock velocity is 20 km/s as intimated by Referee #2. To illustrate this, and this is an extreme case, imagine a jet moving at 100 km/s, the density of which is much lower than its stationary (as seen by the observer) surroundings. At the terminal shock, the jet material is effectively hitting a wall, i.e., its denser surroundings. The shock while strong, i.e., of high Mach number, will be approximately stationary as seen by the observer. Of course this is extreme but it illustrates the point. Moreover one has to be careful when using the apparent relative lengths of the two lobes as a way of checking which side is propagating faster *now* than the other. Bipolar jets can be asymmetrical in density, velocity, etc., and evolve differently in time, so the “history” of one side can be very different than the other, even if they look similar now. A related point is that one has to be careful in equating dynamical timescales, determined by using length over velocity, into evolutionary ones, i.e. dynamical timescales are just indicative.

Here we are just pointing out that the environment of the western side is more dense and its jet is moving more slowly and depositing energy into the western cloud.

In Figure 1, the North direction in the upper-left corner is incorrect. The scale length of 1000 au in the lower-right corner should be shorter. It is not clear how the authors pinpoint the location of the central protostar and determine its coordinates. It would be useful to mention the coordinates of the protostar to be clear for the readers.

Thank you to Referee #2 for pointing out these mistakes. Yes the arrow in the original Figure 1 was pointing the wrong way – it should have been 14° counter-clockwise from straight up, not clockwise! A more accurate au-scale bar has also now been included. The coordinates of HH211-mm, which were known from previous mm interferometer observations, have been added to the supplementary text with an appropriate reference.

BTW, I could not find any information about the angular resolution for each filter. May I suggest the authors to list the angular resolution for each filter, e.g., in Table 1?

This information has now been added to the filter table.